# Clinical Utility of Strain Imaging in Assessment of Myocardial Fibrosis

**DOI:** 10.3390/jcm12030743

**Published:** 2023-01-17

**Authors:** Lang Gao, Li Zhang, Zisang Zhang, Yixia Lin, Mengmeng Ji, Qing He, Mingxing Xie, Yuman Li

**Affiliations:** 1Department of Ultrasound Medicine, Union Hospital, Tongji Medical College, Huazhong University of Science and Technology, Wuhan 430022, China; 2Hubei Province Clinical Research Center for Medical Imaging, Wuhan 430022, China; 3Hubei Province Key Laboratory of Molecular Imaging, Wuhan 430022, China; 4Shenzhen Huazhong University of Science and Technology Research Institute, Shenzhen 518057, China; 5Tongji Medical College and Wuhan National Laboratory for Optoelectronics, Huazhong University of Science and Technology, Wuhan 430022, China

**Keywords:** myocardial fibrosis, strain imaging, clinical utility

## Abstract

Myocardial fibrosis (MF) is a non-reversible process that occurs following acute or chronic myocardial damage. MF worsens myocardial deformation, remodels the heart and raises myocardial stiffness, and is a crucial pathological manifestation in patients with end-stage cardiovascular diseases and closely related to cardiac adverse events. Therefore, early quantitative analysis of MF plays an important role in risk stratification, clinical decision, and improvement in prognosis. With the advent and development of strain imaging modalities in recent years, MF may be detected early in cardiovascular diseases. This review summarizes the clinical usefulness of strain imaging techniques in the non-invasive assessment of MF.

## 1. Introduction

Myocardial fibrosis (MF) represents the excessive deposition and disarrangement of fibrillary collagen in the myocardial extracellular matrix following acute or chronic myocardial damage, which is a common pathological manifestation in patients with end-stage cardiovascular diseases [1,2]. This can result in pathological cardiac remodeling, abnormal cardiac morphology and anatomy, reduced myocardial compliance and myocardial systolic and/or diastolic dysfunction [3,4,5]. Extracellular matrix provides structural basis and prevents myocardial rupture to maintain physiological conditions and is very important for wound healing and tissue regeneration. While continuous and excessive tissue injury contributes to the disproportionate and excess deposition of fibrosis, this disrupts the myocardial architecture and affects the course and prognosis of patients [2,6]. MF development and progression is closely associated with adverse clinical events in a variety of cardiovascular diseases such as heart failure (HF), dilated cardiomyopathy (DCM), hypertrophic cardiomyopathy (HCM), myocardial infarction (MI), valvular heart disease, diabetic cardiomyopathy, arrhythmia and peripheral arterial disease [7,8]. MF quantification serves as an alternative marker for cardiac function, degree of cardiac remodeling and ventricular wall stiffness [9], and has crucial clinical value for risk stratification, clinical decision making, determining the timing of intervention for anti-MF therapy and improving prognosis in patients with cardiovascular disease.

Currently, the main examination modalities for detecting the presence of MF include endomyocardial biopsy, magnetic resonance imaging and echocardiography. Endomyocardial biopsy, an examination that directly visualizes myocardial tissues, is the gold-standard method to detect and quantify MF [7,10]. However, the invasive examination means that a major clinical limitation of endomyocardial biopsy is the risk of complications and sampling error [11]. Late gadolinium enhancement (LGE), T1 mapping and extracellular volume (ECV) fraction assessed by cardiac magnetic resonance (CMR) are currently recognized as the non-invasive reference standard for diagnosing focal and diffuse MF [4,12,13,14,15]. However, CMR examination with special sequences is time-consuming, high-cost, not suitable for patients with gadolinium allergy, renal dysfunction, metal device implantation and, hence, precluding its widespread adoption in the clinical realm [16]. Thus, there is an urgent need to explore other invaluable imaging techniques for accurate, non-invasive, and convenient assessment of MF. MF, acute myocyte injury from a toxin/therapy, acute myocardial ischemia or chest deformity can result in abnormal strain. However, myocardial strain has been demonstrated to correlate with the extent of MF, providing valuable prognostic information in patients with HF, cardiomyopathy, and valvular heart disease [4,17,18]. CMR is the first non-invasive imaging technique to measure myocardial strain. A variety of CMR-based myocardial strain imaging modalities have emerged, e.g., feature tracking (FT), myocardial tagging, strain encoding, of which FT is based on conventional cine CMR and myocardial strain can be obtained without additional imaging and complex post-processing [19]. Echocardiography is the most commonly used tool for assessing myocardial strain. Several recent studies have shown that both two-dimensional speckle tracking echocardiography (2D-STE) and three-dimensional speckle tracking echocardiography (3D-STE) developed in recent years may allow us to accurately evaluate the MF [20,21,22]. Herein, we review the clinical usefulness of strain imaging in evaluation of MF.

## 2. Strain Imaging

Myocardial strain is the percent change in myocardial length relative to the baseline length, and it represents the degree of myocardial deformation and changes over time during the cardiac cycle [23,24,25]. In 1973, Mirsky et al. proposed the concept of strain (ε), which allowed quantitative analysis of myocardial mechanics [26]. Myocardial strain was first measured by CMR tagging in 1988 [27]. In 1998, tissue Doppler imaging (TDI) was introduced to measure one-dimensional myocardial strain and became the first echocardiographic technique to measure myocardial deformation [28]. Two-dimensional speckle tracking echocardiography was developed and applied in 2000 [29,30]. Three-dimensional speckle tracking echocardiography developed over the last decade based on 2D-STE and three-dimensional echocardiography; it can objectively and accurately analyze myocardial strain in three-dimensional datasets.

### 2.1. Cardiac Magnetic Resonance

CMR is recommended for clinical research and practice to non-invasively quantify cardiac function and MF, as CMR can comprehensively assess cardiac morphological structure, function, myocardial perfusion, viability and MF, with high soft tissue resolution [31]. At present, CMR-based tissue tracking imaging techniques include FT, myocardial tagging, strain encoding [19]. The underlying principle of CMR-FT algorithms is similar to speckle tracking echocardiography (STE), where the block-matching approach is the basis for CMR-FT to recognize patterns of features along the blood cavity–myocardial interface in CMR images. We can define the small square windows around these anatomic features on the first image, and then search the most matching image pattern and tracking them during the cardiac cycle in successive images [32,33,34]. Myocardial strain can be obtained using CMR-FT with unrestricted large fields of view, high contrast to noise, and signal-to-noise ratios. In addition, myocardial fiber deformation is quantitatively analyzed without demanding post-processing by parameters such as strain, strain rate, and torsion to further assess myocardial function and MF [33]. Previous studies have reported that CMR-FT strain is highly reproducible and significantly correlated with CMR-LGE [19,35]. Myocardial tagging is the first technique and a current gold standard for strain measurements. It is based on dedicated pulse sequences and induces regional myocardial saturation to facilitate tracking the regions of interest during the cardiac cycle. This approach has yet to be used widely in clinical realms because of the time-consuming nature of additional acquisition and post-processing [36]. Introduced in the early 2000s [37], strain encoding, combined with out-of-plane phase-encoded gradient, encodes strain information into color images to facilitate analysis of abnormal myocardial deformation. However, it is compromised by the low spatial resolution and additional imaging of specialized pulse sequences [19,35].

### 2.2. Speckle Tracking Echocardiography 

Echocardiographic modalities such as STE and TDI are commonly used to the evaluation of myocardial strain, where STE can be used for quantifying global and regional myocardial function non-invasively and objectively by tracking myocardial speckles in a selected cardiac cycle, with the advantages of angle independency, feasibility, fast and high reproducibility [20,38]. STE strain imaging is an advanced tool for quantifying myocardial function and plays an important role in the assessment of MF in many cardiovascular diseases such as HF, cardiomyopathy and valvular heart diseases [24,39,40].

STE is a pioneer in assessing two-dimensional strain. The main advantages of 2D-STE strain measurements in evaluating myocardial function in various diseases have been widely reported and 2D-STE allows for the evaluation of MF in a subclinical period of different cardiovascular diseases [29,41,42,43]. Nevertheless, Myocardial fibers are arranged in complex and contract in different directions simultaneously, while the myocardial motion and mechanics are three-dimensional phenomena [44]. Two-dimensional speckle tracking echocardiography, an imaging technique, works in the two-dimensional planes of tomographic images with the inherent limitations of tracking failure due to out-of-plane speckle motion, apex-to-base foreshortening in the apical views, and the neglect left ventricular torsion, resulting in a less accurate assessment of myocardial deformation that may affect MF evaluation [45]. More recently, three-dimensional myocardial data sets are obtained in the full-volume mode of the apical views by 3D-STE. Three-dimensional speckle tracking echocardiography is based on a speckle tracking pattern-matching algorithm and is a promising technique that is valuable in overcoming the intrinsic restrictions of two-dimensional strain analysis, offering comprehensive and accurate quantitation of the myocardial geometry, global and regional function from a single three-dimensional acquisition. Meanwhile, 3D-STE also has limitations such as low spatial and temporal resolution which affect the assessment of myocardial deformation [20,21,24,44]. STE and CMR-FT strain analysis software automatically delineates endomyocardial boundaries throughout the cardiac cycle; it tracks the motion of the ventricular wall, so as to obtain the myocardial deformation indexes such as global, regional strain and twist (Figure 1, Figure 2, Figure 3 and Figure 4).

## 3. Strain Imaging in Evaluation of MF

### 3.1. Heart Failure

MF is the underlying pathological manifestation of various cardiac diseases, causing HF and cardiac dysfunction. Historic studies have shown that left ventricular ejection fraction (LVEF), the commonly used clinical indicator of cardiac function, fails to identify subtle changes in left ventricular function caused by MF in the early stage. MF remains present in patients with subclinical HF despite normal LVEF [4,13,46]. Therefore, researchers are increasingly interested in employing the analysis of MF as an objective and early marker of cardiac decompensation.

STE strain has a good accuracy in detecting early myocardial structural abnormalities and predicting prognosis in HF patients. Lisi et al. applied 2D-STE to measure left ventricular global longitudinal strain (GLS), right atrial longitudinal strain and right ventricular free wall longitudinal strain (RVFWLS) to investigate the accuracy of right ventricular strain in predicting MF. Their observations showed that the above strain indices were decreased in participants with advanced HF compared with healthy subjects (RVFWLS −15.3 ± 4.7% vs. −29.7 ± 4.7%, *p* < 0.0001). RVFWLS was independently correlated with MF (R^2^ = 0.64, *p* < 0.0001), and had the highest accuracy in diagnosing MF (AUC = 0.87; 95% CI 0.80–0.94). RVFWLS was the major determinant of right ventricular MF and the independent predictor of poor prognosis [18]. In HF patients, chronic elevated left ventricular filling pressure provides a nidus for left atrial progressive damage and MF [47,48]. The same group of authors assessed LA reservoir strain by 2D-STE and left atrial fibrosis by histopathology. Peak longitudinal atrial strain (PLAS) was closely associated with left atrial fibrosis (R = −0.88, *p* < 0.0001) and New York Heart Association class (R = −0.64, *p* < 0.0001). PALS was the strongest predictor of left atrial fibrosis in multivariate regression analysis (β = −0.91, *p* < 0.001) [48]. Thus, 2D-STE strains are of great value in evaluating MF and predicting adverse outcomes in HF patients. Tian et al. explored which strain parameter was best for predicting right ventricular MF by 2D-STE and 3D-STE and found that traditional parameters of right ventricular function, two-dimensional RVFWLS and three-dimensional RVFWLS were reduced in severe MF patients compared with mild and moderate right ventricular MF subjects, and three-dimensional RVFWLS associated best with the extent of right ventricular MF detected histologically (r = −0.72 vs. −0.21 to −0.53, *p* < 0.05; cut-off value = −9.46%, 83% sensitivity, 99% specificity). Three-dimensional speckle tracking echocardiography may be the most powerful imaging technique for accurately predicting the degree of right ventricular MF in end-stage HF patients [21].

In the 2013 full version of AHA/ACC HF guidelines, the potential of CMR to provide comprehensive and accurate assessment about cardiac function, myocardial perfusion and MF, that might assist with the diagnosis of HF and evaluation of HF prognosis was highlighted [31,49]. In a study by Erley et al., CMR strains were closely related to MF measured with CMR-LGE, and CMR-FT global circumferential strain (GCS) has high accuracy and reproducibility in reflecting MF. However, 2D-STE GLS was not significantly related to CMR-LGE, which may be caused by the particular subject populations, prevalence and severity of strain abnormalities, and measurement methods [19]. Shenoy et al. investigated the long-term prognostic value of CMR-FT-derived GLS in a large cohort study. In contrast to the recipients with CMR-GLS above the median, the incidences of major adverse cardiac events (MACE, including cardiovascular mortality, HF hospitalization, MI or stroke) or all-cause mortality were higher in recipients with CMR-GLS below the median. CMR-GLS had an independent predictive value for MF severity [50]. Kammerlander et al. followed HF subjects with preserved LVEF for 38 ± 29 months and obtained the GLS, GCS, global radial strain (GRS) and strain rates to investigate the correlation between CMR-FT strain and diffuse MF from CMR-ECV. The results indicated that the CMR-GLS was associated with CMR-ECV and related to MACE [51]. Current findings regarding the evaluation of MF in patients with HF using strain imaging are presented in Table 1. The above studies showed that STE and CMR strain indexes (RVFWLS, PLAS, GCS or GLS) were able to accurately detect and predict the degree of MF in HF patients and emerged as surrogate parameters of LGE [18,19,50]. Strain imaging was better than ejection fraction and remaining conventional parameters, and independently related to MACE [21,48,50,51].

### 3.2. Dilated Cardiomyopathy

DCM is a common cause of heart transplantation, one of the commonest inherited diseases of heart muscle with biventricular or left ventricular dilatation and deterioration, without significant coronary artery disease and abnormal chamber load (valve disease, hypertension) [52,53]. The estimated prevalences of DCM are reported 40:100,000 and the annual incidence is 7:100,000 [54]. MF is a common pathological manifestation in DCM patients and has been shown to correlate with impairing cardiac function and high mortality [20,55].

Advanced strain imaging modalities (such as STE and CMR-FT) have the potential to analyze the presence and extent of MF before DCM develops [56]. Chimura et al. followed symptomatic DCM participants for 3.8 ± 2.5 years in a longitudinal, retrospective study where they found the significant difference in Two-dimensional GLS (−8.3 ± 3.7% vs. −9.8 ± 4.9%, *p* < 0.05) between participants with LGE (LGE+) and without LGE (LGE-). Only GLS was independently associated with left ventricular remodeling (over CMR-LGE) in multivariable logistic analysis (OR = 0.76, 95% CI 0.67–0.86; cut-off value ≤ −8.5%; both *p* < 0.05) and could be used as an indicator for risk stratification to assess prognosis [57]. In a prospective study with DCM participants who underwent both 2D-STE and CMR-LGE imaging, the parameters of left ventricular GLS, global work index, global work efficiency and global constructive work were derived from 2D-STE pressure–strain loop technique, and showed that the above indicators were reduced in LGE+ participants against those LGE- and remained independent predictors of left ventricular MF [58]. Two-dimensional speckle tracking echocardiography may predict left ventricular MF well, but can 3D-STE strain approximate a histologic biopsy of myocardium? Wang et al. compared STE with pathologic biopsy to evaluate the accuracy and effectiveness of two-dimensional and three-dimensional STE strain indices in detecting the degree of left ventricular MF. Their findings showed that the left ventricular three-dimensional GLS, GRS, tangential strain and two-dimensional GLS were lower in severe MF patients versus patients with mild and moderate MF, where three-dimensional GLS was strongly associated with MF (r = 0.72, *p* < 0.001). The three-dimensional GLS was the robust surrogate marker for assessing the extent of MF in DCM patients with HF (AUC = 0.86, *p* < 0.001, cut-off value = −9.7%) [20]. More recently, Chi et al. applied 3D-STE to explore the correlation between MF degree and left ventricular twist and indicated that the left ventricular global torsion and twist angle were significantly reduced in the DCM group compared with the healthy control group and were closely related to the degree of MF (r = −0.841, −0.828, respectively; both *p* < 0.05). Three-dimensional speckle tracking echocardiography enabled accurate quantification of left ventricular twist dynamics to indirectly reflect the MF conditions in patients with DCM [59].

A retrospective study by Azuma et al. explored MF in non-ischemic DCM patients by CMR-FT versus ECV and investigated the relationship between myocardial strain and ECV. Their data demonstrated that left ventricular longitudinal strain, radial strain, and circumferential strain (−10.2 ± 3.78% vs. −19.8 ± 4.30%, 22.7 ± 10.3% vs. 30.3 ± 18.2%, −6.47 ± 1.89% vs. −9.52 ± 5.15%, respectively; all *p* < 0.01) were significantly lower and ECV (0.30 ± 0.07% vs. 0.28 ± 0.06%, *p* < 0.01) was significantly higher in the DCM group than in healthy controls. CMR-derived radial strain and circumferential strain were significantly correlated with ECV [60]. Buss et al. followed non-ischemic DCM participants for 5.3 years and reported that the mean longitudinal strain (cut-off value > −10%) and GLS (cut-off value > −12.5%) measured by CMR-FT were independent prognostic indexes surpassing LGE mass. CMR-FT longitudinal strain was a survival predictor in DCM (HR = 5.4, *p* < 0.01) and provided additional information on risk stratification [61]. In conclusion, CMR-FT strain can be used as a non-invasive imaging marker for MF detection without contrast media. Table 2 presents the current findings regarding the evaluation of MF in DCM patients using strain imaging. Many studies indicated that strain parameters (GLS, GCS, global constructive work, left ventricular twist) play an important role in diagnosis and prediction of MF, risk stratification and prognostic evaluation of DCM participants [20,58,60]; they were more accurate than other imaging data such as left ventricular stroke volume index [20], left ventricular end-diastolic and end-systolic volumes, LVEF (except for one study [58]), left atrial volume [57], left atrial diameter [59], and LGE mass [61].

### 3.3. Hypertrophic Cardiomyopathy

HCM is defined as a genetic cardiomyopathy with left ventricular wall thickness ≥ 15 mm measured by any imaging method (CMR, echocardiography, or computed tomography, etc.) or unexplained left ventricular wall thickness ≥ 13 mm in first-degree relatives of HCM patients in the absence of load condition anomaly, with a prevalence of 0.2% in the US population and 0.08% in China [24,62,63]. Pathophysiologically, HCM is characterized by cardiomyocyte hypertrophy, disarrangement, and the existence of MF. Many studies have suggested that MF remains a crucial contributor to arrhythmias and mortality in patients with HCM [64,65,66]. The assessment of the MF can offer incremental value for risk stratification and prognostication of HCM patients. Strain imaging may assist with the detection of decreased cardiac function and identification of non-MF and MF segments in HCM patients [24,67].

Many previous studies used strain imaging against pathological specimens or CMR-LGE to detect fibrosis, testing their hypothesis. Popović et al. sought to assess the relationship between MF and segmental strains using CMR-LGE and 2D-STE in HCM patients with preserved LVEF. The results showed that the mean longitudinal strain was associated with the extent of MF (r = 0.46, *p* < 0.005) and the number of myocardial segments with MF (r = 0.47, *p* < 0.005). Both left ventricular wall thickness and MF were predictive factors of depressed segmental longitudinal strain (*p* < 0.005). Interestingly, they also observed the decreased longitudinal, radial, and circumferential strains in HCM patients without MF. In patients with HCM, MF was correlated with reduced longitudinal strain [68]. Saito et al. enrolled HCM patients and revealed that GLS was significantly lower in LGE+ patients than in LGE- patients (−11.8 ± 2.8% vs. −15.0 ± 1.7%, *p* < 0.001). GLS appears to be an independent predictor of the extent of LGE in multivariate analysis (standard coefficient = 0.627, *p* < 0.001). Two-dimensional speckle tracking echocardiography might provide valuable information on MF and MACE (GLS cut-off value = −12.9%) in HCM patients with preserved ejection fraction [69]. The conclusions of another original research by Galli et al. were in line with the aforementioned studies [70]. Put together, these studies validated the results that 2D-STE strain may assess MF and be regarded as a prognostic index in patients with HCM. Pagourelias et al. investigated HCM and hypertension patients who underwent 2D-STE, 3D-STE and CMR-LGE. They demonstrated that the absolute values of two-dimensional longitudinal strain and circumferential strain were significantly higher than those of three-dimensional derivatives, and the correlation between two-dimensional and three-dimensional strain was susceptible to the extent of myocardial hypertrophy and cardiac morphology (R^2^ = 0.66, 0.50, respectively; *p* ≤ 0.001). Moreover, Two-dimensional segmental longitudinal strain (AUC = 0.78, 95% CI 0.75–0.81, *p* < 0.0005) was the optimal deformation parameter for detecting subtle MF in HCM patients [71]. Urbano-Moral et al. aimed at comparing 3D-STE and CMR-LGE to depict left ventricular global and regional myocardial mechanics of HCM. Intriguingly, they observed that the degree of myocardial hypertrophy and MF were the main factors of myocardial mechanical changes. Moreover, LGE range and three-dimensional GLS damage were correlated with the degree of myocardial hypertrophy. Three-dimensional segmental area strain seems to be a sensitive parameter reflecting MF in HCM [72]. Another prospective study published recently indicated that the three-dimensional GLS was moderately correlated with CMR-LGE (r = 0.465, *p* = 0.001), and 3D-STE could detect MF determined by CMR-LGE (GLS cut-off value = −15.25%, 84.6% sensitivity, 84.8% specificity) [73]. Song et al. analyzed the relationship between various strain parameters of CMR-FT and histological MF and the incremental value of FT strain to CMR-LGE in patients with symptomatic obstructive HCM, and the results showed that the longitudinal, circumferential, and radial strains of the interventricular septum in the LGE- patient group with increased MF were significantly lower than those without (all *p* < 0.05) [74]. They proved that the above strain parameters were significantly correlated with MF detected by myocardial biopsy (r = 0.32, 0.42, −0.27, respectively; all *p* < 0.005), where longitudinal strain independently correlated with MF and had incremental value over CMR-LGE for detecting MF [74]. Therefore, CMR-FT is an imaging tool that provides important value in quantitative analysis of MF degree and can be used as an imaging marker to detect MF in obstructive HCM patients. CMR-FT strain is able to offer more useful additional information versus CMR-LGE in detecting the extent of MF. The conclusion corroborates the results of another single-center retrospective cohort study published previously (GLS cut-off value ≤ −12.8%, 91% sensitivity, 89% specificity) [75]. Table 3 illustrates current findings regarding the evaluation of MF in HCM patients using strain imaging. The above studies demonstrated that strain imaging could provide good incremental value over the classic LVEF, left ventricular wall thickness, left ventricular mass index, and LGE in detecting and predicting MF and MACE in HCM patients [68,69,70,71,72,73,74,75].

### 3.4. Myocardial Infarction

Ischemic injury following MI elicits the reactive remodeling of the myocardium in which the damaged areas are replaced with permanent fibrotic scars produced by myofibroblasts and fibroblasts to prevent cardiac rupture. However, exaggerated and massive fibrosis causes the progressive reduction in cardiac function, alters the stiffness of the ventricular wall, and ultimately leading to HF [76,77]. Altiok et al. included 93 subjects with first acute MI who underwent the percutaneous coronary intervention in a retrospective study, and applied CMR-LGE to determine their global and regional myocardial scars; they confirmed that 2D-STE and CMR-LGE had similar accuracy in predicting postoperative left ventricular remodeling as well as global left ventricular functional improvement in subjects with acute MI (AUC = 0.806 vs. 0.824, 0.715 vs. 0.729, respectively; both *p* > 0.50). Two-dimensional speckle tracking echocardiography allows effective analysis of the presence and degree of myocardial scar in subjects with MI [78]. Huttin et al. prospectively enrolled 100 subjects with first ST-elevation MI to assess the value of 3D-STE strain in detecting myocardial impairment and microvascular obstruction after acute MI. They reported that all three-dimensional strain indexes in infarcted segments were significantly reduced than those in non-infarcted segments, where the segmental area strain and radial strain were related to increased diagnostic accuracy of transmural LGE (AUC > 0.04, *p* < 0.01). The 3D-STE was a sensitive tool for quantifying the transmural scars [79]. CMR with delayed contrast enhancement (DCE), the current reference standard technique, was used for identifying the location and extent of MF. DCE suggests irreversible myocardial impairment, including focal MF caused by non-ischemic etiology or MI [80,81]. In an observational investigation enrolling 113 patients with ischemic or non-ischemic left ventricular dysfunction and using 3D-STE and CMR-DCE to predict MACE, the three-dimensional GCS and global area strain were significantly reduced in patients with more MACE, and 3D-STE strains were closely related to LVEF. There was no significant difference in DCE between the two groups. Compared to DCE, Three-dimensional speckle tracking echocardiography allowed better identification of patients who develop cardiac-adverse events and early detection of their myocardial injury [80]. Maret et al. aimed to determine whether CMR-FT may accurately assess the myocardial scars, which defined with CMR-LGE after percutaneous coronary intervention in 30 patients with ST-elevation MI. They showed a significant decline in all strain parameters of myocardial segments with scar, of which radial strain was the best strain parameter for detecting transmural myocardial scars (radial strain AUC = 0.89, cut-off value = 38.8%; longitudinal strain AUC = 0.76, cut-off value = −18.5%) [82]. CMR-FT play a robust role in detecting myocardial scar in MI patients.

### 3.5. Aortic Stenosis

Aortic stenosis (AS) is one of the most common types of valvular heart disease in Western countries with the prevalence of 12.4% in the elderly, 3.4% of whom are severe [83,84]. The pressure or volume overload due to valvular heart disease is one of the common causes of diffuse MF. Usually, patients with AS present with raised left ventricular systolic pressure, which further causes concentric compensatory hypertrophy of the left ventricle and ultimately contributes to progressive MF, increased left ventricular stiffness, decreased compliance, and deterioration of diastolic function [85,86,87]. Understanding the interaction between the left ventricular systolic parameters (e.g., LVEF, systolic strain) and the loading conditions is essential for clinical decision making [85,88,89]. Slimani et al. compared 101 subjects with severe AS with healthy controls undergoing 2D-STE to measure GCS, GLS and wall-stress indexes at end-systole. The extent of interstitial MF was significantly aggravated in subjects with decreased preoperative stress–strain parameters (GCS 4.9 ± 4.4% vs. 9.5 ± 6.4%, GLS 3.8 ± 2.9% vs. 8.3 ± 6.3%; both *p* < 0.001). Stress–strain parameters before surgery were the independent predictors of circumferential and longitudinal stress–strain parameters after surgery (β = 0.60, 0.49, respectively; both *p* < 0.001); predicting the degree of postoperative cardiac function recovery and recognizing patients with a large extent of MF [88]. Park et al. revealed the moderate correlation between GLS value assessed by 2D-STE and the degree of histological MF (r = 0.421; *p* = 0.0003) in severe AS patients. Compared to GLS and native T1, ECV was better to predict the extent of MF (R^2^ = 0.35, 0.36, 0.44; Akaike Information Criterion= 66.84, 66.18, 55.8; respectively) [90]. Interstitial MF and ventricular wall hypertrophy are known to be the consequences of pressure overload in patients with AS [91], and STE strain measurements have been used for assessment of MF and myocardial viability. Hoffmann et al. demonstrated that CMR-LGE-derived MF was negatively correlated with left ventricular peak longitudinal strain (r = −0.538, *p* = 0.007), and the cut-off value of peak longitudinal strain (<−11.6%) well- identified MF (LGE > 10%) had a specificity of 75% and a sensitivity of 65% (AUC = 0.69). There was a correlation between STE strain and the degree of MF measured by CMR-LGE [92]. A prospective study involving the marker of replacement MF on CMR in 261 patients with moderate to severe AS underwent STE. Two-dimensional left ventricular GLS indicated great MF discrimination (AUC = 0.74, 95% CI 0.66–0.83, *p* < 0.001) and standardization (Hosmer–Lemeshow χ^2^ = 6.37; *p* = 0.605) and independently related to replacement of MF. In addition, patients with GLS > −15.0% (equivalent to 95% specificity included in MF) had the increased MACE compared to those with LV-GLS < −15.0%. GLS was correlated with the replacement MF (LGE), whereas it is weakly associated with reactive interstitial MF (ECV and native T1). Two-dimensional speckle tracking echocardiography could be used as a good strain imaging technique to assess replacement MF in AS patients [93]. Patients with severe AS usually have pulmonary hypertension (PH), which correlates with poor prognosis [94]. Combined with myocardial biopsy, CMR-FT and two-dimensional echocardiography data from 34 subjects with severe AS, Gumauskiene et al. analyzed the correlation between PH progression and left ventricular MF. In contrast to the subjects without PH, the degree of diffuse MF was significantly raised in subjects with severe AS and PH (6.6 (4.6–8.2)% vs. 12 (10.4–12.7)%, *p* < 0.001). The degree of diffuse MF was associated with lower LVEF, increased pulmonary artery systolic pressure (PASP), increased NT-proBNP, left ventricular GLS and GCS (r = −0.6, 0.6, 0.5, −0.5, −0.5, respectively; all *p* ≤ 0.05). MF > 10% (AUC = 0.95, sensitivity 100%, specificity 84%, *p* < 0.001) and GLS > −15.5% (AUC = 0.86, sensitivity 100%, specificity 82.6%, *p* < 0.001) were independent predictors of PH in patients with severe AS. The combination of histological diffuse MF, GLS, and plasma NT-proBNP offers incremental value for risk stratification and prognosis in severe AS patients [94].

### 3.6. Mitral Regurgitation

In patients with mitral regurgitation (MR), partial left ventricular blood flow ejected into the left atrium during systole and into the left ventricle again during diastole. Chronic volume overload offers a nidus for increased myocardial length and elevated diastolic stress, which contributing to progressive left cardiac dilatation, remodeling, aggravated MF and myocardial dysfunction [4,95,96,97]. The left atrium is the main MR target organ injury and strain imaging allows an objective and accurate quantification of left atrial myocardial deformation and function [98,99]. Her et al. observed the significant association between the global left atrial strain and left atrial fibrosis (r = −0.55, *p* < 0.001), and the two-dimensional strain of left atrium might assist with predicting left atrial fibrosis in mitral valve patients [100]. A study by Cameli et al. investigating 46 severe MR patients applying 2D-STE versus histopathology showed that among the two-dimensional echocardiographic indexes, the diagnostic accuracy of global PLAS in detecting left atrial fibrosis was the highest (AUC = 0.89), and PLAS was closely related to the degree of left atrial fibrosis (r = −0.82, *p* < 0.0001) and was an independent predictor of left atrial fibrosis [101]. Subsequently, Mandoli et al. followed 65 primary severe MR subjects undergoing mitral surgery for 3.7 ± 1 years or 6.8 ± 1 years (event-group and non-event group, respectively) and determined the association of left atrial strain with postoperative prognosis and left atrial fibrosis by myocardial biopsy. PALS and left ventricular GLS were significantly lower in the poor prognosis group (17 ± 8% vs. 28 ± 8%, −17.3 ± 4.1% vs. −20.6 ± 4.4%, respectively; both *p* ≤ 0.01). Moreover, Global PLAS offered useful prognosis information (AUC = 0.78, *p* < 0.01; cut-off value = 21%) and risk stratification information (90 ± 5% (PALS ≥ 21%) vs. 52 ± 9% (PALS < 21%) for event-free survival in 5 years, *p* < 0.0004) for subjects with severe MR. It was an independent predictor of HF and mortality and was negatively correlated with histologically verified left atrial MF (r^2^ = 0.80, 31.9 ± 20.8% (PALS ≥ 21%) vs. 76.6 ± 20.7% (PALS < 21%); *p* < 0.0001). Global PLAS was a crucial imaging marker of left atrial fibrosis [39]. Ischemic MR is susceptible to right ventricular volume and pressure overload, resulting in non-ischemic MF and right ventricular dysfunction [102]. A recent study suggested that RV strains were strongly associated with CMR-derived right ventricular ejection fraction and might serve as the good markers of non-ischemic MF in patients with MR [102]. Huber et al. applied CMR-FT to explore the correlation between left atrial strain and the extent of left atrial fibrofatty remodeling in MR patients, reporting that the impairment of PLAS was observed in MR patients compared with healthy controls (17.0 ± 9.7% vs. 35.6 ± 8.6%, *p* < 0.001). Altered left atrial strain was significantly associated with raised left atrial remodeling, and PLAS was strongly related to the severity of left atrial fibrofatty replacement (r = −0.75, *p* = 0.017) [103]. CMR-FT strain measurement is beneficial to guide treatment strategies and delay disease progression.

Several studies mentioned above have demonstrated the value of strain imaging in clinical practice. In early stages, strain imaging is of great help in identifying subclinical cardiac dysfunction and defining prognosis. Moreover, the functions of cardiac chambers are significantly reduced in patients with advanced cardiac disease due to significant MF, and the application of strain imaging is mainly focused on assessing severity and prognosis, when myocardial strains are the most accurate functional indexes [22,33]. Normal strain values help clinicians to rationally exclude the severe MF. However, strain imaging is an indirect diagnostic tool for MF; decreased strain in individual patients is not necessarily a sign of the presence of MF and strain techniques should always be applied and must be combined with clinical features and other examinations as artifactual factors, or interferences may influence strain results [43,104]. For patients with gadolinium allergy, renal dysfunction, metal device implantation or critical illness, STE should always be applied. For patients with poor echocardiographic quality, CMR-FT should always be applied. For patients with poor image quality, strain imaging adds little to the information from standard imaging [16,32]. Strain techniques are still undergoing development, and further validation is required before it can be recommended as a routine application; strain imaging can currently be used as a complementary method to aid clinical diagnosis and decision making.

## 4. Limitations and Future Directions

This study is limited by the inherent nature of literature review, such as publication bias and selection bias, as well as the articles indexed only in the PubMed and China National Knowledge Infrastructure databases (to avoid low-quality data). It is important to analyze thoroughly the weak aspects of strain imaging techniques to confirm their usability. The additional limitation of the study is that there are not many studies on the assessment of MF using strain imaging modalities; thus, more evidence based on larger sample sizes is needed to demonstrate its clinical utility and to incorporate the results into recommendation statements and clinical protocols.

MF affects the progression of many cardiac diseases (e.g., HF, DCM, ischemic cardiomyopathy and valvular heart diseases) and is an unmet public-health problem. Previously, MF could be identified only by invasive histological evaluation, while new advances in imaging methods have raised the likelihood of noninvasive myocardial tissue characterization in recent years. Conventional echocardiography and CMR indices show a low sensitivity since a decrease in cardiac function may be subtle, and the correlation between strain measures and MF is little-known [105]. Strain assessment isincreasingly being used due to this technique allowing a more precise analysis of subclinical cardiac dysfunction and providing indirect information about MF when LVEF >50% in specific clinical situations [29]. Owing to the irreversible nature of MF processes, early and accurate assessment of MF has an important role in risk stratification and prognosis of patients. Our review has interesting findings that can have crucial implications for clinical practice. A large number of encouraging results offered clinical standpoints on the use of strain techniques to improve patient management and guide clinical decision making. The encouraging results derived from different studies provide clinical perspectives on the use of these techniques for guidance in clinical decision making and improvement in the management of patients with AS. Future applications, namely machine learning algorithms (i.e., artificial intelligence), will automate myocardial function assessment and are expected to be a promising technology to improve diagnostic accuracy and prognosis, accelerating clinical decision making.

## 5. Summary

MF is a critical pathological manifestation of chronic cardiac dysfunction and HF. Early identification and assessment of MF play a crucial role in risk stratification, therapeutic strategies, timing of intervention, and improvement in prognosis in patients with various cardiovascular diseases. Myocardial strain imaging has attracted attention because of its non-invasive, convenient, efficient, and accurate advantages. Many investigations have reported that strain indexes may act as the best surrogate marker of MF and were independent predictors of adverse cardiovascular events in patients with cardiovascular disease. With the continuous development and improvement in myocardial strain imaging technology, the combination of different imaging modalities with complementary advantages to accurately evaluate the degree of MF and changes in cardiac function offers a reference basis for clinical management and treatment in patients with cardiovascular disease.

## Figures and Tables

**Figure 1 jcm-12-00743-f001:**
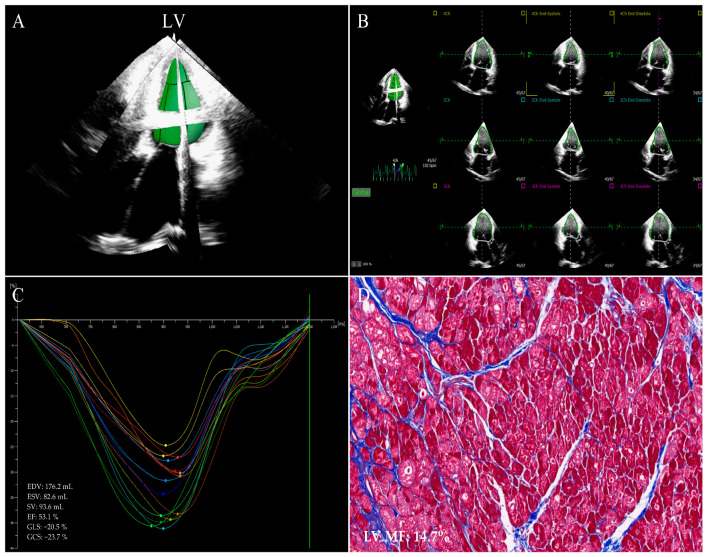
LV images of 3D−STE strain in a HF patient who underwent the heart transplantation. (**A**) Three−dimensional LV reconstruction image. (**B**) LV endocardial border outlining and tracking. (**C**) LV global strain. Different colored lines represent the strain curves of different myocardial segments of LV. (**D**) Example of LV myocardial samples with Masson’s staining obtained from the explanted heart immediately after transplantation (original magnification ×200). LV: left ventricular; MF: myocardial fibrosis; 3D−STE: three−dimensional speckle tracking echocardiography; HF: heart failure.

**Figure 2 jcm-12-00743-f002:**
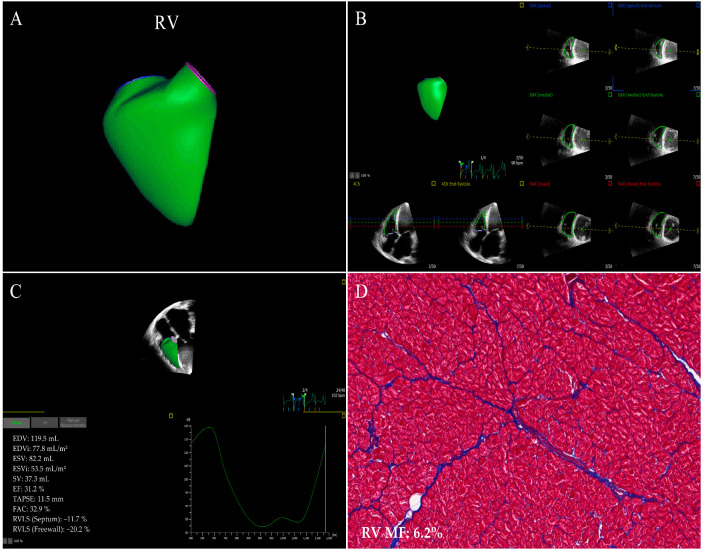
RV images of 3D−STE strain in a HF patient who underwent the heart transplantation. (**A**) Three−dimensional RV reconstruction image. (**B**) RV endocardial border outlining and tracking. (**C**) longitudinal strain of RV septum and free wall. (**D**) Example of RV myocardial samples stained with Masson’s staining acquired from the explanted heart (original magnification ×200). RV: right ventricular; other abbreviations as in Figure 1.

**Figure 3 jcm-12-00743-f003:**
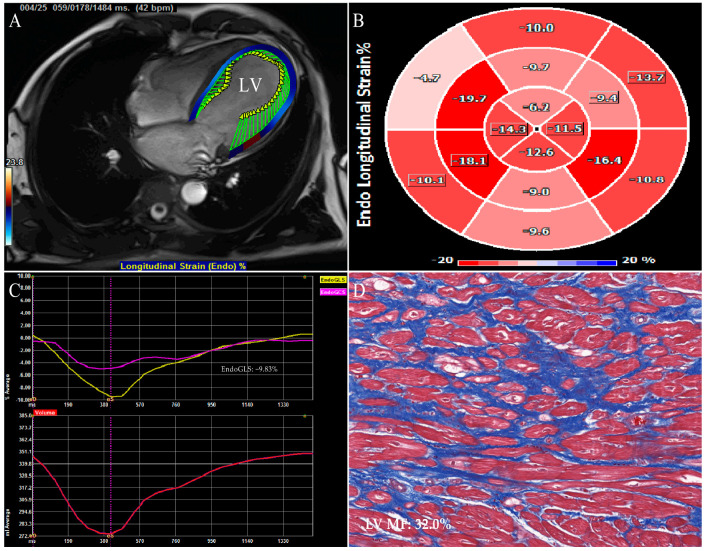
CMR−FT strain images of left ventricle in a DCM patient who underwent the heart transplantation. (**A**) LV endocardial border outlining and tracking. (**B**) Bulls−eye plot of LV segmental longitudinal strain. (**C**) LV global longitudinal strain (yellow curve) and global circumferential strain (pink curve). (**D**) Example of LV myocardial samples stained with Masson’s staining (original magnification ×200). CMR−FT: cardiac magnetic resonance feature tracking; DCM: dilated cardiomyopathy; other abbreviations as in Figure 1 and Figure 2.

**Figure 4 jcm-12-00743-f004:**
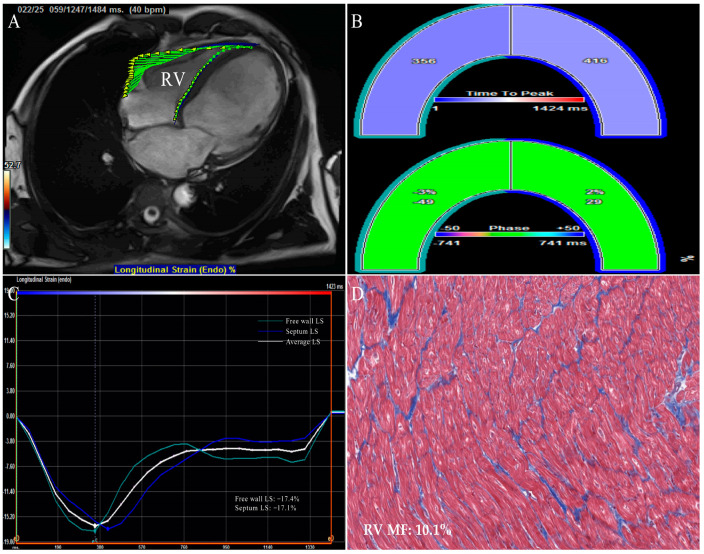
CMR−FT strain images of right ventricle in a DCM patient who underwent the heart transplantation. (**A**) RV endocardial border outlining and tracking. (**B**) Time to peak RV longitudinal strain and phase. (**C**) RV longitudinal strain. (**D**) Example of RV myocardial samples stained with Masson’s staining (original magnification ×200). The abbreviations as in Figure 1, Figure 2 and Figure 3.

**Table 1 jcm-12-00743-t001:** Evaluation of MF in HF patients using strain imaging.

References	Year	Sample Size	Age (Years)	Strain Imaging Technique	Reference Standard	Main Findings	Confidence Level
Lisi et al. [18]	2015	27	53.7 ± 4.6 ^a^ 51.8 ± 6.7 ^b^	2D-STE	Histopathology	RVFWLS was independently correlated with MF. RVFWLS was the major determinant of RV MF and the independent predictor of poor prognosis.	*p* < 0.0001
Lisi et al. [48]	2022	48	51.2 ± 8.1	2D-STE	Histopathology	PLAS was closely associated with left atrial fibrosis and was the strongest predictor of left atrial fibrosis.	*p* < 0.001
Tian et al. [21]	2021	105	44 ± 16	2D-STE, 3D-STE	Histopathology	Three-dimensional RVFWLS associated best with the extent of RV MF.	*p* < 0.05
Erley et al. [19]	2019	50	51 ± 9	2D-STE, CMR-FT	CMR-LGE	CMR-GCS was closely related to MF measured with CMR-LGE.	*p* = 0.041
Shenoy et al. [50]	2020	152	54 ± 15	CMR-FT	CMR-LGE	CMR-GLS had an independent predictive value for MF severity.	*p* < 0.001
Kammer-lander et al. [51]	2020	206	71 ± 8	CMR-FT	ECV	The CMR-GLS was associated with CMR-ECV and related to MACE.	*p* < 0.05

^a^ Patients with HF; ^b^ Healthy controls. MF: myocardial fibrosis; HF: heart failure; 2D-STE: two-dimensional speckle tracking echocardiography; RVFWLS: right ventricular free wall longitudinal strain; RV: right ventricular; PLAS: peak longitudinal atrial strain; 3D-STE: three-dimensional speckle tracking echocardiography; CMR: cardiac magnetic resonance; FT: feature tracking; LGE: late gadolinium enhancement; GCS: global circumferential strain; GLS: global longitudinal strain; ECV: extracellular volume; MACE: major adverse cardiac events.

**Table 2 jcm-12-00743-t002:** Evaluation of MF in DCM patients using strain imaging.

References	Year	Sample Size	Age (Years)	Strain Imaging Technique	Reference Standard	Main Findings	Confidence Level
Chimura et al. [57]	2017	179	61 ± 15	2D-STE	CMR-LGE	Two-dimensional GLS was independently associated with LV remodeling and could be used as an indicator for risk stratification to assess prognosis in participants with DCM.	*p* < 0.05
Cui et al. [58]	2021	57	43.9 ± 12.7	2D-STE	CMR-LGE	Two-dimensional speckle tracking echocardiography indicators were reduced in LGE+ participants against with those LGE- and remained independent predictors of LV MF.	*p* < 0.05
Wang et al. [20]	2021	75	44 ± 15	2D-STE, 3D-STE	Histopathology	Three-dimensional GLS, GRS, tangential strain and two-dimensional GLS were lower in severe MF patients versus patients with mild and moderate MF, where three-dimensional GLS strongly associated with MF. Three-dimensional GLS strongly associated with MF.	*p* < 0.001
Chi et al. [59]	2022	38	50.77 ± 9.45 ^a^ 51.33 ± 10.33 ^b^	3D-STE	CMR-LGE	The three-dimensional global torsion and twist angle were significantly reduced in DCM group and were closely related to the degree of MF.	*p* < 0.05
Azuma et al. [60]	2020	57	61 ± 12	CMR-FT	ECV	LV strains were significantly lower and ECV was significantly higher in the DCM group than in healthy controls. CMR radial strain and circumferential strain were significantly correlated with ECV.	*p* < 0.01
Buss et al. [61]	2015	210	52 ± 15	CMR-FT	CMR-LGE	The longitudinal strains were independent prognostic indexes surpassing LGE mass and ejection fraction. CMR-FT longitudinal strain was a survival predictor in DCM and provided additional information on risk stratification.	*p* < 0.01

^a^ Patients with DCM; ^b^ Healthy controls. DCM: dilated cardiomyopathy; LV: left ventricular; GRS: global radial strain; other abbreviations as in Table 1.

**Table 3 jcm-12-00743-t003:** Evaluation of MF in HCM patients using strain imaging.

References	Year	Sample Size	Age (Years)	Strain Imaging Technique	Reference Standard	Main Findings	Confidence Level
Popović et al. [68]	2008	39	42 ± 14 ^a^ 44 ± 16 ^b^	2D-STE	CMR-LGE	Both LV wall thickness and MF were predictive factors of depressed longitudinal strain.	*p* < 0.003
Saito et al. [69]	2012	48	63 ± 14	2D-STE	CMR-LGE	GLS was significantly lower in LGE+ patients than in LGE- patients. GLS appears to be an independent predictor of the extent of LGE.	*p* < 0.001
Galli et al. [70]	2019	82	58 ± 14	2D-STE	CMR-LGE	The HCM patients had significantly decreased global constructive work, which was related to MF assessed by LGE.	*p* = 0.04
Pagourelias et al. [71]	2020	40	54.1 ± 14.3	2D-STE, 3D-STE	CMR-LGE	The correlation between two-dimensional and three-dimensional strain was susceptible to the extent of myocardial hypertrophy and cardiac morphology. Two-dimensional segmental longitudinal strain was the optimal deformation parameter for detecting subtle MF.	*p* < 0.0005
Urbano-Moral et al. [72]	2014	59	61 ± 12	3D-STE	CMR-LGE	The degree of myocardial hypertrophy and MF were the main factors of myocardial mechanical changes, LGE range and three-dimensional GLS damage were correlated with the degree of myocardial hypertrophy.	*p* ≤ 0.01
Spartera et al. [73]	2017	50	52 ± 18.5	3D-STE	CMR-LGE	Three-dimensional GLS was moderately correlated with CMR-LGE, and 3D-STE could detect MF determined by CMR-LGE.	*p* = 0.001
Song et al. [74]	2022	123	46.25 ± 14.17	CMR-FT	Histopathology	The interventricular strains were significantly correlated with MF, where longitudinal strain independently correlated with MF.	*p* ≤ 0.01
Bogarapu et al. [75]	2016	29	13.5 ± 6.1	CMR-FT	CMR-LGE	GLS, GCS, GRS and strain rate were reduced in pediatric patients with HCM and MF. GLS ≤ −12.8% has high accuracy for detecting MF.	*p* = 0.0007

^a^ Patients with HCM with MF; ^b^ Patients with HCM without MF. GCS: global circumferential strain; other abbreviations as in Table 1 and Table 2.

## Data Availability

Not applicable.

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
