# Peer review of "Clinical Utility of Strain Imaging in Assessment of Myocardial Fibrosis"

_jcm, 2023, doi:10.3390/jcm12030743_

Round 1

Reviewer 1 Report

The introduction is the clearest aspect of the manuscript. Somewhere the authors need to acknowledge that MF is not the only cause of altered strain. For example, acute myocyte injury from a toxin/therapy, chest deformity or acute ischemia can result in abnormal strain. 

Who is the intended audience for the review?  Most of the data that are related to the specific clinical conditions are already well known and have been reviewed for cardiologists.  The very detailed presentation of all the many clinical studies will distract the general audience; this would need to be summarized in either Tables or in the text.  One should discuss when the strain data, whether by 2D-STE or MRI techniques, are truly useful. Presenting the statistical significance does not specifically address the potential utility. Studies that demonstrate reduced strain in patients with obvious myocardial dysfunction by standard imaging should be carefully reviewed as to the population being studied.  The level of confidence that strain data provide in each clinical situation should be summarized. 

The overview of strain imaging (2.0) is clear and would be appropriate for a general audience.  If the intended reader is a cardiologist interpreting cardiac imaging, they would already be familiar with this information and this section could be shortened.  Figure 1 is attractive but difficult to read/interpret – much too small.

Tables 1,2 and 3 are nice summaries of the results of the many studies.  What is missing is a subsequent discussion as to how and when strain imaging should be a routine part of the analysis of myocardial function/MF in each of these clinical conditions.  For example, in the section on heart failure could you present the data regarding the potential role of strain imaging to assess prognosis? Is it more accurate than other known imaging data such as LVEF? This should be discussed for both imaging techniques (CMR and 2D-STE).  It is not very informative to know that myocardial strain is reduced when there is other evidence of myocardial dysfunction.

The first sentence of the summary should include the additional word “chronic” cardiac dysfunction.

Author Response

Response to Reviewer 1 Comments

Point 1: The introduction is the clearest aspect of the manuscript. Somewhere the authors need to acknowledge that MF is not the only cause of altered strain. For example, acute myocyte injury from a toxin/therapy, chest deformity or acute ischemia can result in abnormal strain.

 Response 1:

Thank you very much for your valuable review. We have added the relevant descriptions to the introduction. (Lines 57-58)

Point 2: Who is the intended audience for the review?  Most of the data that are related to the specific clinical conditions are already well known and have been reviewed for cardiologists.  The very detailed presentation of all the many clinical studies will distract the general audience; this would need to be summarized in either Tables or in the text.  One should discuss when the strain data, whether by 2D-STE or MRI techniques, are truly useful. Presenting the statistical significance does not specifically address the potential utility. Studies that demonstrate reduced strain in patients with obvious myocardial dysfunction by standard imaging should be carefully reviewed as to the population being studied.  The level of confidence that strain data provide in each clinical situation should be summarized.

Response 2:

Thank you very much for your suggestion. We consider the intended readers of this review are cardiologists and general audiences interested in myocardial strain imaging and MF, so we want to describe relevant information as clearly and accurately as possible. We reckon that strain data, whether by 2D-STE or MRI techniques, are useful to assess the degree of MF, especially in patients with gadolinium allergy, renal dysfunction and metal device implantation. According to your valuable comments, we have revised and summarized the detailed presentation of the many clinical studies and added the confidence level of strain data in the tables. (Lines 172-173, 181-182, 187-188, 205, 209, 219-220, 236, 242, 245, 250-251, 257, 264, 271, 285-286, 302, 309, 317-318, 326, 336, 353-354)

Point 3: The overview of strain imaging (2.0) is clear and would be appropriate for a general audience.  If the intended reader is a cardiologist interpreting cardiac imaging, they would already be familiar with this information and this section could be shortened.  Figure 1 is attractive but difficult to read/interpret – much too small.

Response 3:

Thank you for your comment. We have appropriately shortened the overview section of strain imaging (2.0) and adjusted Figure 1 for improving the readability in the revised manuscript. (Lines 74-75, Line 141)

Point 4: Tables 1,2 and 3 are nice summaries of the results of the many studies.  What is missing is a subsequent discussion as to how and when strain imaging should be a routine part of the analysis of myocardial function/MF in each of these clinical conditions.  For example, in the section on heart failure could you present the data regarding the potential role of strain imaging to assess prognosis? Is it more accurate than other known imaging data such as LVEF? This should be discussed for both imaging techniques (CMR and 2D-STE).  It is not very informative to know that myocardial strain is reduced when there is other evidence of myocardial dysfunction.

Response 4:

We thank the reviewer for these valuable comments. We have added the subsequent discussions (diagnosis, cut-off value, risk stratification, prognosis, etc.) of tables 1,2 and 3 in the revised manuscript. (Lines 192-193, 214-218, 240, 255-256, 272-273, 278-284, 312-314, 347-352)

Point 5: The first sentence of the summary should include the additional word “chronic” cardiac dysfunction.

Response 5:

Thank you very much for your careful review. We have included the additional word “chronic” in first sentence of the summary. (Line 512)

Once again, we thank this reviewer for the extremely insightful and constructive comments and critiques.

Reviewer 2 Report

This is an interesting study regarding the clinical usefulness of strain imaging techniques in the non-invasive assessment of myocardial fibrosis. Early analysis of myocardial fibrosis has an important role in risk stratification and prognosis of patients. I consider that the article is valuable and has interesting findings that can have important implications for clinical practice. In order to improve the quality of the manuscript, I have some suggestions:

1. Please include the details of the broader impacts of the study made, addressing the future scope and topics that are important.

2. Please consider including the limitations of the study.

Author Response

Response to Reviewer 2 Comments

Point 1: This is an interesting study regarding the clinical usefulness of strain imaging techniques in the non-invasive assessment of myocardial fibrosis. Early analysis of myocardial fibrosis has an important role in risk stratification and prognosis of patients. I consider that the article is valuable and has interesting findings that can have important implications for clinical practice. In order to improve the quality of the manuscript, I have some suggestions:

Please include the details of the broader impacts of the study made, addressing the future scope and topics that are important.

Response 1:

Thank you very much for your confirmation and suggestion. We have added a section that includes the details of the broader impacts of the study made, addressing the future scope and the important topics. (Lines 490-509)

Point 2: Please consider including the limitations of the study.

Response 2:

We thank the reviewer for the valuable comments. We have added a part of the limitations of the study. (Lines 481-489)

Once again, we appreciate the expertise and insight of the reviewer and the very helpful comments and questions.  

Reviewer 3 Report

This is a nice review article to address potential usefulness of 2D/3D echocardiography (2D/3DE) speckle tracking derived strain or CMR feature tracking derived strain for the evaluation of myocardial fibrosis. I have a few minor comments that should be addressed by the authors.

1.     Are there any studies to fail to show the significance of 2D/3DE strain (CMR feature tracking strain) for evaluating myocardial fibrosis? I think that there are some publication biases. If you find negative results, you will describe these results. This could be the neutral.

2.     Are there any publications showing that 2D/3DE strain (CMR strain) has incremental values over CMR-LGE or ECV for the association of the outcome? If you find lots, you will summarize the results.

3.     Figures are not appropriate. Please consider showing strain results and estimation of myocardial fibrosis (LGE or ECV) in couple of cases.

Author Response

Response to Reviewer 3 Comments

Point 1: This is a nice review article to address potential usefulness of 2D/3D echocardiography (2D/3DE) speckle tracking derived strain or CMR feature tracking derived strain for the evaluation of myocardial fibrosis. I have a few minor comments that should be addressed by the authors.

  Are there any studies to fail to show the significance of 2D/3DE strain (CMR feature tracking strain) for evaluating myocardial fibrosis? I think that there are some publication biases. If you find negative results, you will describe these results. This could be the neutral.

Response 1:

Thank you very much for your valuable review. We have added the negative results regarding the significance of strain for evaluating myocardial fibrosis in the revised manuscript. (Lines 201-203, 411-413, 427-428)

Point 2: Are there any publications showing that 2D/3DE strain (CMR strain) has incremental values over CMR-LGE or ECV for the association of the outcome? If you find lots, you will summarize the results.

Response 2:

Thank you for your comment. We found a few studies showing the incremental values of 2D/3DE strain (CMR strain) over CMR-LGE or ECV for the association of the outcome and added them to the revised manuscript. (Lines 239, 335-336, 342)

Point 3: Figures are not appropriate. Please consider showing strain results and estimation of myocardial fibrosis (LGE or ECV) in couple of cases.

Response 3:

Thank you very much for your review. The patients of Figures 1 and 2 had undergone the heart transplantation. So, we used the histological confirmation of MF as a reference method, rather than the MF assessed by LGE or ECV. We have adapted Figures 1 and 2 to show strain results and estimates of myocardial fibrosis (histopathological images). (Lines141-148, 151-158)

Once again, we appreciate the insightful and constructive comments and questions of the reviewers. 
